# TLR4-Mediated Recognition of Mouse Polyomavirus Promotes Cancer-Associated Fibroblast-Like Phenotype and Cell Invasiveness

**DOI:** 10.3390/cancers13092076

**Published:** 2021-04-25

**Authors:** Vaclav Janovec, Boris Ryabchenko, Aneta Škarková, Karolína Pokorná, Daniel Rösel, Jan Brábek, Jan Weber, Jitka Forstová, Ivan Hirsch, Sandra Huérfano

**Affiliations:** 1Department of Genetics and Microbiology, Faculty of Science, Charles University, BIOCEV, 25150 Vestec, Czech Republic; vaclav.janovec@natur.cuni.cz (V.J.); boris.ryabchenko@natur.cuni.cz (B.R.); jitka.forstova@natur.cuni.cz (J.F.); huerfano@natur.cuni.cz (S.H.); 2IOCB Gilead Research Center, Institute of Organic Chemistry and Biochemistry of the Czech Academy of Sciences, 16000 Prague, Czech Republic; karolina.pokorna@uochb.cas.cz (K.P.); jan.weber@uochb.cas.cz (J.W.); 3Department of Cell Biology, Faculty of Science, Charles University, BIOCEV, 25150 Vestec, Czech Republic; aneta.skarkova@natur.cuni.cz (A.Š.); daniel.rosel@natur.cuni.cz (D.R.); jan.brabek@natur.cuni.cz (J.B.); 4Institute of Molecular Genetics of the Czech Academy of Sciences, 14220 Prague, Czech Republic

**Keywords:** mouse polyomavirus, MPyV, mouse fibroblasts, CAF, TLR4, IL-6, spheroid invasiveness

## Abstract

**Simple Summary:**

Mouse polyomavirus (MPyV) is widely used as a model for cancer development studies. In many preceding studies, its tumorigenic potential was attributed to a virus protein called middle T antigen (MT), which possesses a transforming ability through activation of cell-associated tyrosine kinases, resulting in increased cell growth. Here, we studied the effects of the innate immune responses triggered by MPyV in mouse fibroblasts. We found that recognition of MPyV by Toll-like receptor 4 (TLR4), a sensor of the innate immunity system, induces the production of interleukin 6 (IL-6) and other cytokines without inhibiting virus multiplication. The cytokine microenvironment changed the phenotype of adjacent noninfected fibroblasts toward the cancer-associated fibroblast (CAF)-like phenotype associated with increased chemokine production and invasiveness. Thus, our data indicate that MPyV contributes to the CAF-like phenotype in mouse fibroblasts via a TLR4-driven inflammatory response.

**Abstract:**

The tumorigenic potential of mouse polyomavirus (MPyV) has been studied for decades in cell culture models and has been mainly attributed to nonstructural middle T antigen (MT), which acts as a scaffold signal adaptor, activates Src tyrosine kinases, and possesses transforming ability. We hypothesized that MPyV could also transform mouse cells independent of MT via a Toll-like receptor 4 (TLR4)-mediated inflammatory mechanism. To this end, we investigated the interaction of MPyV with TLR4 in mouse embryonic fibroblasts (MEFs) and 3T6 cells, resulting in secretion of interleukin 6 (IL-6), independent of active viral replication. TLR4 colocalized with MPyV capsid protein VP1 in MEFs. Neither TLR4 activation nor recombinant IL-6 inhibited MPyV replication in MEFs and 3T6 cells. MPyV induced STAT3 phosphorylation through both direct and MT-dependent and indirect and TLR4/IL-6-dependent mechanisms. We demonstrate that uninfected mouse fibroblasts exposed to the cytokine environment from MPyV-infected fibroblasts upregulated the expressions of MCP-1, CCL-5, and α-SMA. Moreover, the cytokine microenvironment increased the invasiveness of MEFs and CT26 carcinoma cells. Collectively, TLR4 recognition of MPyV induces a cytokine environment that promotes the cancer-associated fibroblast (CAF)-like phenotype in noninfected fibroblasts and increases cell invasiveness.

## 1. Introduction

Polyomaviruses are DNA tumor viruses that are widely spread in nature; they infect mammals and birds. Serological studies and DNA sequencing have shown that polyomaviruses are highly prevalent in the human population. To date, 14 human polyomaviruses have been described. Among them, BK polyomavirus (BKPyV), JC polyomavirus (JCPyV), *Trichodysplasia spinulosa* polyomavirus, and Merkel cell virus (MCPyV) cause disease in humans. The oncogenic potential of the polyomaviruses was demonstrated several decades ago for simian polyomavirus virus 40 and for mouse polyomavirus (MPyV), but among the human viruses, MCPyV was not clearly linked to the development of skin cancer only in 2007 [1,2,3].

Currently, MPyV continues to be the best model to study tumorigenesis since, for the human MCPyV infection of primary dermal fibroblasts, specific conditions are needed [4]. MPyV induces a variety of tumors when inoculated in newborn mice [5]. Its tumorigenic potential has been studied for many years in cell culture models and was attributed mainly to the viral nonstructural middle T antigen (MT), which possesses high transforming ability [6,7]. MT is inserted into the endoplasmic reticulum membrane through the KDEL sequence at the C-terminus; from there, it migrates to the cell periphery [8]. MT localized on both endosomal and plasma membranes act as a scaffold signal adaptor, which activates Src tyrosine kinases [9,10], PI3K [11], PLC-γ1 [12], and PKB/Akt [13], and other cellular kinases. Modulation of cellular signaling by MT creates favorable conditions for viral replication, which can, under some circumstances, be reverted to cellular transformation [14]. Besides MPyV T antigens, host genetic variations and the immune response play an important role in the susceptibility to MPyV tumorigenesis [15].

In general, activation of the innate immune responses inhibits the initial viral spread and leads to proper activation of the adaptive immune response [16,17]. Velluipallai et al. [18] showed that Toll-like receptor (TLR) 4 is the key mediator of the cytokine response, which governs susceptibility to tumor development during MPyV infection. Particularly, the authors showed that polymorphism in TLR4 drives the differences in susceptibility to tumor induction by MPyV in the resistant mouse strain C57BR/cdJ (BR) in contrast to the susceptible mouse strain PERA/Ei (PEA). Antigen-presenting cells (APCs) from the BR strain recognize MPyV through TLR4 and produce IL-12, which induces the TH1 T cell response, whereas APCs from the PEA strain produce IL-10, which favors the TH2 cell response [18,19].

Many cytokines such as IL-6 or TNF-α are produced during wound healing in mice [20], and it was shown that cytokines like TGF-β and TNF-α increase replication of human polyomaviruses in vitro [21,22]. Moreover, the triggering of chronic inflammation by persistent viral infections has been clearly demonstrated [23]. Chronic inflammation is associated with hyperproduction of cytokines that support growth, promote immunosuppression of T lymphocytes, and are commonly present in the cancer environment [24,25]. For example, IL-6 overproduction by stromal cells supports tumor growth through STAT3 activation [26], promotes cancer-associated fibroblast (CAF)-induced cancer invasion, and was recently shown to promote resistance of cancer cells to therapy, proving its broad pro-tumorigenic role [27,28].

Several lines of evidence link persistent polyomavirus infection with inflammation-associated pathologies in humans. Results obtained on the mouse model support the role of polyomaviruses in chronic inflammation. MT-transformed endothelial cells (MT-ECs) continually secrete IL-6, which serves as both an autocrine and paracrine growth factor [29,30]. Consistently, MT-EC-transplanted mice treated with anti-IL-6 antibodies showed a lower frequency of metastases [29], supporting the notion that the cytokine environment drives tumorigenesis and metastasis formation [24]. Here, using a cell culture model, we addressed the question of whether TLR4 recognition of MPyV induces cytokine secretion in mouse fibroblasts, which alters fibroblast phenotype and promotes cell invasiveness in vitro.

## 2. Materials and Methods

### 2.1. Cell Lines and Virus

We grew 3T6 (ATCC; CCL-96) mouse fibroblasts, mouse embryonic fibroblasts (MEFs) (ATCC; CCL-13), and mouse CT26 colon carcinoma cells at 37 °C in a 5% CO_2_-air humidified incubator using Dulbecco′s modified Eagle′s medium (DMEM; Sigma-Aldrich, Saint Louis, MO, USA) supplemented with 10% fetal bovine serum (Thermo Fisher Scientific, Waltham, MA, USA) and Pen/strep (Thermo Fisher Scientific). MPyV (BG strain) was purified from infected 3T6 cells and titrated as previously described [31]. Cell-free medium from MPyV-infected cells (MPyV-CM) was prepared with 10% sucrose cushion ultracentrifugation. The absence of MPyV virions in MPyV-CM was tested by flow cytometry. Total exosome isolation from MPyV-CM was performed with Total Exosome Isolation Reagent obtained from Thermo Fisher Scientific.

### 2.2. Inhibitors, Antibodies, and Reagents

TLR4 inhibitor CLI-095, TLR4 agonist lipopolysaccharide (LPS), TLR4 antagonist LPS-RS, JAK1/2 inhibitor ruxolitinib, and bafilomycin A1 were obtained from InvivoGen (San Diego, CA, USA). Recombinant mouse IL-6 was purchased from PeproTech (Cranbury, NJ, USA). siRNA targeting mouse TLR4 and control siRNA were purchased GE Healthcare Dharmacon (Lafayette, CO, USA). Lipofectamine RNAiMAX Transfection Reagent was obtained from Thermo Fisher Scientific. Mouse monoclonal antibody targeting mouse α-SMA was obtained from R&D systems (Minneapolis, MN, USA). Mouse monoclonal antibody anti-MPyV T antigens, mouse monoclonal antibody targeting MPyV LT antigen and rabbit polyclonal antibody targeting VP1 were used as described previously [31,32,33]. Mouse monoclonal antibodies targeting mouse TLR4 were purchased from Santa Cruz (Dallas, TX, USA) and Abcam (Cambridge, UK). Rabbit polyclonal antibody targeting mouse Rab11 and mouse monoclonal antibody targeting mouse ALIX were obtained from Santa Cruz. Mouse monoclonal IgG1 antibody targeting GAPDH was obtained from Thermo Fischer Scientific. Rabbit monoclonal antibody targeting STAT3, rabbit monoclonal antibody targeting phospho-STAT3 (Tyr705), and rabbit monoclonal antibody targeting phospho-STAT-3 (Ser727) were obtained from Cell Signaling Technology (Danvers, MA, USA). Goat anti-mouse monoclonal antibody conjugated with Alexa Fluor 488, goat anti-mouse monoclonal antibody conjugated with Alexa Fluor 546, and goat-anti-mouse conjugated with APC were purchased from Thermo Fisher Scientific. Donkey anti-rabbit monoclonal antibody conjugated with PE was obtained from Biolegend (San Diego, CA, USA).

### 2.3. Cell Stimulation In Vitro and Viral Infection

MEF and 3T6 cells were stimulated with 1 µg/mL LPS (Invivogen, San Diego, CA, USA). To inhibit TLR4 function, cells were pretreated with 10 µg/mL LPS-RS (Invivogen) or 1 µM CLI-095 (Invivogen) 1 h before LPS or MPyV addition. The 3T6 cells or MEFs were infected with MPyV at a multiplicity of infection (MOI) = 5 plaque-forming units (PFU)/cell (when not specified otherwise), diluted in serum-free medium for 1 h at 37 °C. After virus adsorption, complete DMEM medium with 10% FBS was added.

### 2.4. Immunofluorescence Staining and Confocal Microscopy

Cells were washed with PBS and then fixed with 4% formaldehyde in PBS for 15 min. Cells were then permeabilized with 0.5% Triton X-100 in PBS for 5 min and washed 3× with PBS. Cells were blocked with 1% BSA for 1 hour and incubated with rabbit polyclonal antibody to VP1 [31], mouse monoclonal anti-TLR4 antibody (Abcam), and rabbit polyclonal anti-Rab11 antibody (Santa Cruz). Mouse monoclonal antibody targeting mouse α-SMA (R&D systems) was used for α-SMA staining. We used goat anti-mouse conjugated with Alexa Fluor 488 and goat anti-rabbit conjugated with Alexa Fluor 546 as secondary antibodies. Images were obtained with a TCS SP8 confocal microscope (Leica, Wetzlar, Germany). Pearson′s correlation coefficient (PCC) was calculated as previously described [34] using ImageJ [35] with the JaCoP plugin [36].

### 2.5. Measurement of IL-6 Secretion

The amount of IL-6 produced by 3T6 or MEFs was measured in cell-free supernatants using mouse ELISA kits (Mabtech, Stockholm, Sweden). Briefly, 3T6 or MEFs were stimulated by LPS or infected with MPyV, and cell-free supernatants were collected 24 h post-stimulation or -infection. Supernatants were centrifuged and analyzed according to the manufacturer′s protocol.

### 2.6. Determination of IL-6, CCL2 /MCP-1, SDF-1, α-SMA, and IP-10 Expression

Total cellular RNA was isolated using a High Pure RNA Isolation Kit (Roche). cDNA was synthesized using an iScript cDNA Synthesis Kit (Biorad, Hercules, CA, USA). The mRNA of interest was amplified with a LightCycler 480 SYBR Green I Masterkit (Roche, Basel, Switzerland) using the following primers: IL-6: forward: 5′-CGTGGAAATGAGAAAAGAGTTGTGC-3′ and reverse 5′-CAGGTAGCTATGGTACTCCAGAAG-3′; CCL2/MCP1: forward: 5′-AAGACTGAATGGCTGGATGGC-3′; SDF-1: forward: 5′-AACTCGCTCCTCCCTCTTCG-3′ and reverse 5′-GGGAAGAGTTTACCGTCAGGT-3′; α-SMA: forward: 5′-CTACGAACTGCCTGACGGG-3′ and reverse 5′- GCTGTTATAGGTGGTTTCGTGG-3′; IP-10: forward: 5′-TGCAGGATGATGGTCAAGCC-3′ and reverse 5′-CACTTGAGCGAGGACTCAGA-3′; and Ppia: forward: 5′-AAGACTGAATGGCTGGATGGC-3′ and reverse 5′-CATTCCTGGACCCAAAACGC-3′. Relative expression levels were calculated using the 2^−ΔΔCT^ method. Mouse peptidylprolyl isomerase A (Ppia) was used as the endogenous control.

### 2.7. TLR4 Silencing Using siRNA

MEFs were plated to 70% confluency 1 day before transfection. Then, siRNA against TLR4 or control siRNA was transfected using RNAiMAX (Thermo Fisher Scientific). TLR4 expression was analyzed 48 h post-transfection by Western blot.

### 2.8. Determination of STAT3 Phosphorylation, Alpha-SMA, ALIX, and TLR4 by Immunoblotting

Total STAT3 and alpha-SMA in the whole cell lysate of MEFs were determined by Western blot using of rabbit monoclonal antibody targeting STAT3 (Cell Signaling using Technology) and mouse monoclonal antibody targeting mouse α-SMA (R&D systems). Phosphorylation of STAT3 in the whole cell lysate of MEF cells was analyzed by Western blot using rabbit monoclonal antibody targeting phospho-STAT3 (Tyr705) (Cell Signaling Technology) and rabbit monoclonal antibody targeting phospho-STAT-3 (Ser727) (Cell Signaling Technology). ALIX was determined in exosomes isolated from MPyV-CM by Western blot using mouse monoclonal antibody targeting ALIX (Santa Cruz). After incubation with the appropriate horseradish peroxidase-conjugated secondary antibody, the membranes were washed, and the protein bands were detected with SuperSignal West Femto Maximum Sensitivity Substrate (Thermo Fisher Scientific). Densitometric analyses were performed using an Amersham Imager 600 (GE Healthcare Life Science, Marlborough, MA, USA). Band intensities were normalized to GAPDH detected by mouse monoclonal antibody targeting GAPDH (Thermo Fisher Scientific). The whole un-cropped images of Western blots are shown in Appendix A.

### 2.9. Determination of STAT3 Phosphorylation and MPyV Infection of MEFs by Flow Cytometry

To determine MPyV-infected cells by flow cytometry, MEFs or 3T6 cells were trypsinized and fixed with 4% formaldehyde in PBS for 15 min. Cells were then permeabilized with 0.1% Triton X-100 in PBS for 5 minutes and washed 3× with PBS. Cells were incubated with the mouse monoclonal anti-MPyV T-antigen common region. For flow cytometry analysis of phospho-STAT3 (Tyr705), cells were fixed in 4% formaldehyde for 10 min, permeabilized by 90% methanol for 30 min, and stained by rabbit monoclonal antibody targeting phospho-STAT3 (Tyr705) (Cell Signaling Technology). We used goat anti-mouse conjugated with APC (Thermo Fisher Scientific) and donkey anti-rabbit conjugated with PE (Biolegend). Live/dead cell discrimination was performed using a Zombie Green™ Fixable Viability Kit (BioLegend). Samples were analyzed using a BD LSR FORTESSA cytometer (BD Biosciences, San Jose, CA, USA), and data were processed using FLOWJO software (Treestar, San Carlos, CA, USA).

### 2.10. Cell Invasiveness Assay

Cells were grown in micro-mold 3D Petri Dish Microtissues (Merck, Darmstadt, Germany) according to the manufacturer′s protocol for 2 days to obtain multicellular spheroids of defined size. Then, the spheroids were transferred and embedded into a 3D collagen matrix (final composition 1 mg/mL rat tail collagen, 1 × RPMI medium, 15 mM HEPES, 1% fetal bovine serum, and 50 μg/mL gentamicin) and overlaid with either conditioned medium from MPyV-infected MEFs (MPyV-CM) or control conditioned medium from noninfected MEFs (MOCK-CM). In the case of inhibitor treatment, 1 µM ruxolitinib or an equivalent amount of DMSO was added to the overlaying conditioned medium. Images of spheroids were taken immediately after embedding into collagen (0 h) and after invasion (48 h) using a Nikon ECLIPSE TE2000-S microscope. The area of the spheroids before and after invasion was measured using FiJi software, and the relative invasion index was calculated. The data were statistically analyzed in GraphPad Prism 6 (GraphPad Software, La Jolla, CA, USA) using one-way ANOVA. The presented data are summarized from 3 independent biological replicates, and a minimum of 5 spheroids per condition and replicate were analyzed.

### 2.11. Statistical Analysis

Quantitative variables are expressed as the mean  ±  standard error of the mean (SEM). To compare the levels of cytokine production and transcription of mRNA by MEFs and 3T6 cells, we used the Mann–Whitney test and a two-tailed *t*-test. For flow cytometry analyses, we used a two-tailed *t*-test. For the cell invasiveness assay, we used one-way ANOVA. For TLR4/VP1 colocalization, we used Kolmogorov–Smirnov test. Data were analyzed with GraphPad Prism 6 (GraphPad Software). A *p*-value of ≤0.05 was considered significant.

## 3. Results

### 3.1. Mouse Embryonic Fibroblasts and 3T6 Cells Recognize MPyV through TLR4 and Secrete IL-6

The ability of mouse macrophages to recognize MPyV particles by TLR4 and secrete various cytokines [18] led us to investigate whether MEFs recognize MPyV infection in a similar way. We measured IL-6 secretion by MEFs infected with MPyV (MOI = 5) or, as a control, in MEFs stimulated with TLR4 agonist LPS. To further confirm the specific TLR4 activation, TLR4 was inhibited by pretreatment with LPS-RS (TLR4 antagonist) or by CLI-095 (TLR4 inhibitor) (Figure 1a). Both MPyV-infected MEFs and LPS-stimulated MEFs produced significant amounts of IL-6. Pretreatment of MEFs exposed to MPyV or LPS with TRL4 antagonist LPS-RS and TLR4 inhibitor CLI-095 led to the inhibition of IL-6 secretion (Figure 1a). Next, we investigated whether the multiplicity of infection influences IL-6 secretion and whether 3T6 fibroblast cells are able to recognize MPyV. The 3T6 cells were infected with various MOI, and IL-6 secretion was measured until 7 days post-infection (Figure 1b). IL-6 secretion by 3T6 cells was time-dependent and correlated with the level of MOI. To verify the hypothesis that TLR4 recognizes the virus but does not require the viral nuclear phase (transcription/replication), we used bafilomycin A1. Bafilomycin A1 inhibits the vacuolar H^+^-ATPase preventing endosomal acidification. In MPyV infection, this drug causes viral accumulation in early endosomes, thereby preventing further viral trafficking. We previously showed that bafilomycin A1 reduces MPyV replication in 3T6 cells by 90% [32]. Here, cells were treated with bafilomycin and infected with MPyV or, as a control, with LPS. We observed that IL-6 production was inhibited (Figure 1c) neither in the MPyV infection nor in the LPS control (Figure 1c). Thus, active viral replication is not necessary for IL-6 production. Collectively, MEFs and 3T6 cells recognized MPyV infection, and the TLR4 recognition of MPyV was dose-dependent, although it was independent of active viral replication.

### 3.2. TLR4 Colocalizes with MPyV Capsid Protein VP1 in MEF

We showed earlier that MPyV particles enter cells in monopinocytic vesicles that are further sorted through the endosomal system and localized to various endosomal components including Rab5-, Rab7-, and Rab11-positive endosomes [33]. Here, we infected MEFs and analyzed whether the TLR4 receptor also colocalizes with the virus in the endosomes using confocal microscopy. We observed colocalization of MPyV (detected by VP1 capsid protein staining) and TLR4 as early as 2 h post-infection (2 hpi) at a high virus input (MOI = 10). As a result, large clusters of TL4-VP1 were detected (Figure 2a). At the same time point, we followed MEFs infected with low virus input (MOI = 1) and observed only sporadic colocalization of MPyV (VP1) and TLR4. Conversely, very strong colocalization of TLR4 and VP1 at low virus input (MOI = 1) was observed 4 days post-infection (4 dpi) (Figure 2b), as quantified using Pearson′s correlation coefficient (PCC) by including pixels that co-localize between both channels. PCC values can range from +1 (perfect correlation) to −1 (perfect anti-correlation) (Figure 2c). The colocalization between VP1 and TLR4 was greater at 4 dpi than at 2 hpi. In addition, since Rab11 endosomes, which are a part of the route of sorting of MPyV, play an essential role in the trafficking of TLR4 during activation, we followed mutual colocalization of Rab11 and TLR4 in MPyV-infected MEFs and MOCK-treated MEFs. We observed colocalization between TLR4 and Rab11 in infected MEFs (Figure 2d). Thus, our data showed that the virus colocalizes with TLR4 in the endosomal compartments and that TLR4 is internalized to Rab11-positive endosomal compartments during MPyV infection of MEFs.

### 3.3. Neither TLR4 Activation Nor Recombinant IL-6 Inhibits MPyV Replication in MEFs and 3T6 Cells

TLR4 activation in MEFs or 3T6 cells led to the production of proinflammatory cytokine (Figure 1). In primary human hepatocytes, TLR4 stimulation inhibits HBV replication [37]. We tested whether pharmacologic targeting of TLR4 signaling with CLI-095 affects MPyV replication in MEFs determined by the expression of T antigens (Figure 3a). Inhibition of TLR4 signaling in MEFs by CLI-095 did not lead to the decrease in T-antigen-positive MEFs. The same result was obtained with 3T6 cells (Appendix A). It was previously reported that human polyomavirus BKPyV is highly resistant to proinflammatory cytokines [38]. Thus, we stimulated TLR4 in 3T6 cells with LPS or exposed the cells to recombinant IL-6 and IFN-γ 24 h prior to MPyV infection. Consistent with previously published data [39], only IFN-γ pretreatment inhibited MPyV infection in 3T6 cells (Figure 3b). Then, we confirmed our results with siRNA targeting TLR4 in MEFs (Figure 3c,d). TLR4 silencing reduced IL-6 production in MEFs induced by both MPyV and LPS (Figure 3d), while the quantity of large T-antigen (LT) was unchanged (Figure 3c). Altogether, our results demonstrated the nonprotective role of TLR4 activation against MPyV infection in MEFs and 3T6 cells.

### 3.4. MPyV Induces STAT3 Phosphorylation via IL-6

Since TLR4 activation did not inhibit MPyV replication in MEFs and 3T6 cells, we decided to test whether TLR4-mediated IL-6 secretion affects cell signaling. Similar to MT, which promotes STAT3 activation [40], IL-6 also induces STAT3 phosphorylation [41]. First, we showed that T-antigen-positive MEFs contain elevated levels of phosphorylated STAT3 at Y705 compared to T-antigen-negative MEFs using PhosphoFlow cytometry (Figure 4a). Then, we analyzed the effect of the cytokine environment induced by TLR4 recognition of MPyV on STAT3 phosphorylation in noninfected MEFs. We prepared conditioned media (CM) from MPyV-infected fibroblasts (MPyV-CM) or control media from MOCK-infected fibroblasts (MOCK-CM) by ultracentrifugation in a 10% sucrose cushion to separate infectious viral particles (present in pellets) from cytokines (present in the supernatant). We confirmed the presence of IL-6 (600–800 pg/mL) in MPyV-CM by ELISA. Because IL-6 is known to be associated with and transferred by exosomes [42,43], and we detected an exosome marker ALIX in the exosomal pellet of MPyV-CM by Western blot (Appendix A), we quantified IL-6 level in exosomes isolated from MPyV-CM. IL-6 was not detected in the isolated exosomes by ELISA (Appendix A). No virus infectivity was detected after MPyV-CM treatment of MEFs. We assessed whether cytokines, including IL-6 present in MPyV-CM, induce STAT3 phosphorylation at Y705 and S727 by means of Western blot. MPyV-CM significantly elevated the phosphorylation of STAT3 in MEFs, whereas MOCK-CM did not (Figure 4b). To elucidate whether the cytokine environment is responsible for the induction of STAT3 phosphorylation in MEFs, we used JAK1/2 inhibitor ruxolitinib and anti-IL-6 antibodies. Both anti-IL6 antibodies and ruxolitinib decreased the phosphorylation of STAT3 at Y705 (Figure 4c). Thus, our results disclosed a dual mechanism of STAT3 activation due to the presence of MT in MPyV-infected MEFs and IL-6 present in the supernatant.

### 3.5. MPyV Infection Induces Cytokine Environment That Changes MEF Phenotype

Quiescent fibroblasts exposed to CM from cancer organoids alter the phenotype to CAFs and support tumor growth [44]. Thus, we tested whether MPyV-CM treatment can alter the MEF phenotype. First, we analyzed the expression of alpha-smooth muscle actin (α-SMA), which serves as a marker of CAFs [45]. We found that α-SMA expression was elevated in MPyV-CM-treated MEFs compared with MOCK-CM-treated MEFs (Figure 5a). Then, we analyzed the α-SMA structure using confocal microscopy. α-SMA formed vigorous bundles in MPyV-CM-treated MEFs compared with MOCK-CM-treated MEFs. Thus, not only higher expression but also morphological changes in α-SMA were observed in MPyV-CM-treated MEFs (Figure 5b). Conversion of normal fibroblast to CAFs is associated with the secretion of various cytokines/chemokines, which support the cancer environment [46]. As such, we analyzed the mRNA levels of SDF-1, MCP-1, CCL-5, and α-SMA in MPyV-CM-treated MEFs (Figure 5c). Except for SDF-1, we detected high levels of transcription of CAF-associated cytokines in MPyV-CM-treated MEFs.

### 3.6. MPyV Infection in Fibroblasts Establishes Cytokine Environment That Supports Cell Invasiveness

CAFs promote the invasive behavior of cancer cells by both direct and indirect mechanisms [45]. It was also reported that MT-expressing endothelial cells produce IL-6 and induce recruitment of host endothelial cells in vivo [29]. Thus, we tested whether cytokines produced by MPyV-infected fibroblasts can also alter cell invasiveness. To this end, we employed spheroid invasion assays in 3D collagen gels, which are used to mimic in vitro cell dissemination in 3D conditions [47]. We cultured noninfected MEFs (Figure 6a) and mouse colon carcinoma cell line CT26 (Figure 6b) as spheroids and supplemented them with MPyV-CM or MOCK-CM. MPyV-CM significantly increased the spheroid invasion in both cell types. Next, we tested whether JAK1/2 inhibitor ruxolitinib is able to block spheroid invasion. Ruxolitinib pretreatment decreased spheroid invasion in MPyV-CM-treated MEFs (Figure 6c) but not in CT26 cells (Figure 6d), pointing to a difference between primary embryonic fibroblast and cancer-derived cell-line invasiveness. Collectively, MPyV-infected fibroblasts secrete cytokines that affect cellular motility. In particular, the ability of ruxolitinib to significantly inhibit cell invasion shows that IL-6 or other JAK1/2-activating cytokines contribute to the increased invasive behavior of MEFs.

## 4. Discussion

In this study, we found that TLR4-mediated recognition of MPyV in MEF and 3T6 cells led to the formation of a cytokine environment that did not affect MPyV replication. Our data suggest a novel mechanism, where MPyV replication promotes a CAF-like phenotype in adjacent noninfected fibroblasts and increases the cell invasiveness of primary embryonic cells and cancer-derived cells in vitro.

TLR4-mediated recognition of MPyV in MEF and 3T6 cells led to the secretion of proinflammatory cytokine IL-6 and was virus-dose-dependent. We did not observe any cytokine production 24 h post-infection from 3T6 cells when a low viral input was used. However, IL-6 production increased progressively over time, suggesting that uncontrolled viral propagation surpasses the TLR4 activation threshold [48]. When we blocked MPyV infection with bafilomycin A1, IL-6 secretion was not reduced. This is consistent with a previous report showing that MPyV virus-like particles containing only VP1 capsid protein activated the innate immune response in mice [19]. Altogether, we confirmed by TLR4 antagonist LPS-RS, TLR4 inhibitor CLI-095, and TLR4 siRNA targeting that TLR4 is a key mediator of the cytokine response in MPyV-infected MEFs. Our results are consistent with previously published data on TLR2/TLR4 double-knockout macrophages, in which the TLR4 cDNA from BR mice conferred a robust IL-12 response to MPyV [18].

Surprisingly, we did not observe any effect of TLR4 activation or inhibition on MPyV replication in MEFs and 3T6 cells. Recombinant IL-6 did not show any significant effect on MPyV replication in mouse fibroblasts. Similar to our findings, replication of BKPyV is resistant to proinflammatory cytokines [40]. Only IFN-γ was able to inhibit MPyV infection [41]. Silencing of TLR4 reduced the amount of IL-6 produced by MEFs; however, the level of LT antigen was unchanged. We observed TLR4 clusters after MPyV infection that colocalized with Rab11. We previously showed that sorting Rab11-positive recycling endosomes did not represent a productive pathway for the infection of MPyV, and viral particles in them were observed only in a low quantity (14.9 ± 0.9%) [33]. Moreover, MPyV viral particles also localize in Rab5-, Rab7-, Rab11-, LAMP-2-, and caveolin-positive endosomes [33,49]. Our results indicate that trafficking of TLR4 to Rab11 endosomes may be a part of the sorting of the receptor that occurs during TLR4 activation [50]. It has been shown that Rab11 can translocate from recycling endosomes to autophagosomes in response to autophagy induction and assist in the fusion of late endosomes with autophagosomes [51]. Crispr/Cas9 targeting of TLR4 or MEFs from TLR4-knockout mice could be used for specific localization of TLR4 and MPyV in future experiments to decipher the connection of TLR4 with MPyV trafficking.

IL-6 is a multi-functional cytokine that plays roles not only in viral defense but also in the pathogenesis of viral diseases. Exogenous recombinant IL-6 activates STAT3 transcription factor, which is also activated by MPyV MT and is associated with MPyV-induced cell transformation in vitro [42]. MPyV-CM induced strong STAT3 phosphorylation in MEF despite a lower amount of IL-6 in MPyV-CM (600–800 pg/mL) compared to recombinant IL-6 (10 ng/mL). Crosstalk between other cytokines/chemokines and IL-6/STAT3 signaling pathway could be responsible for the MPyV-CM-induced STAT3 phosphorylation. Moreover, exosomes present in MPyV-CM can potentially transfer IL-6 [43], although no IL-6 was detected in the exosome isolated from MPyV-CM by ELISA. More precise analysis of exosomes in a culture medium supplemented with exosome-depleted FBS and more sensitive detection of IL-6 are needed to exclude the role of exosomal IL-6. Altogether, our and other results have shown that MPyV activates STAT3 by phosphorylation directly through MT and indirectly via TLR4-mediated IL-6 production [42].

CAFs represent variable fibroblast populations with several specific effects such as α-SMA expression, cytokine/chemokine secretion, and extracellular matrix remodeling [45]. Soluble factors such as proinflammatory cytokines IL-1-β or IL-6 promote the pro-tumorigenic fibroblast phenotype [44,46]. We found that CM from MPyV-infected MEFs alters normal MEFs to express more α-SMA. Higher α-SMA expression in MEFs was also connected with a higher level of mRNAs encoding CCL5 and MCP-1 chemokines. Several studies reported that CAF-mediated production of CCL5 and MCP1 modulates the cancer environment [52,53]. Our data showed that MPyV replication in MEFs induces a cytokine environment, which promotes the CAF-like phenotype in fibroblasts. MPyV resembles human papillomaviruses that also alter fibroblast phenotype through the IL-6/STAT3 pathway [54,55]. Further investigation is necessary to clarify whether persistent MPyV infection alters the fibroblast phenotype in vivo.

Injection of mice with MT-expressing mouse endothelial cells that secrete soluble factors induced rapid recruitment of host non-transformed endothelial cells into the site of injection [34,52]. Moreover, supernatants from MT-expressing mouse endothelial cells stimulated in vitro invasiveness of cancer cell lines [56]. Thus, we investigated whether the MPyV-induced cytokine environment affects the cell invasiveness of MEFs or the colon carcinoma cell line, CT26. Both MEFs and CT26 migrated more efficiently in the presence of CM from MPyV-infected MEFs, and the migratory effect was inhibited by JAK1/2 inhibitor ruxolitinib in the case of MEFs. However, the lack of ruxolitinib effect on CT26 cell invasiveness suggested that cancer cells are able to circumvent the inhibition of JAK1/2 using an alternative pathway. In addition, the conditioned medium from MPyV-infected MEFs contains several soluble factors, which may have different cellular specificities, and induces both JAK1/2-dependent and -independent increases in cellular invasiveness. This finding is consistent with that of a previous study in which several soluble factors affecting cellular motility were identified in supernatants from MT-expressing endothelial cells [56].

## 5. Conclusions

Here, we demonstrated that mouse fibroblasts recognize viral MPyV particles by TLR4 and created a cytokine environment that is nonprotective against MPyV, increases cellular motility, and changes the phenotype of noninfected fibroblasts toward CAF-like phenotype. Aberrant TLR4 activation in the cancer environment can support tumor growth [57,58]. Thus, active MPyV replication contributes to tumor formation also via a TLR4-driven chronic inflammatory response. Demonstration of the hijacking of the TLR4 pathway in mouse fibroblasts by infecting MPyV illustrates a subtle equilibrium between the antiviral and proviral tumorigenic roles of innate immunity in different cell contexts.

## Figures and Tables

**Figure 1 cancers-13-02076-f001:**
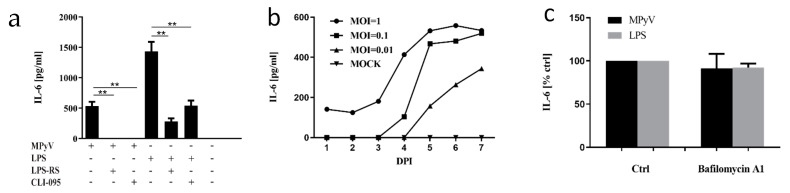
Toll-like receptor 4 (TL4)-mediated recognition of mouse polyomavirus (MPyV) in mouse fibroblasts. (**a**) The production of interleukin (IL)-6 by mouse embryonic fibroblast (MEFs) stimulated with lipopolysaccharide (LPS) or infected with MPyV (multiplicity of infection (MOI) = 5) in the presence of LPS-RS or CLI-095. (**b**) The kinetics of IL-6 secretion by 3T6 cells infected with MPyV with various MOI, 7 days post-infection (DPI). (**c**) The effect of bafilomycin A1 pretreatment on IL-6 secretion by 3T6 cells infected with MPyV (MOI = 5) or stimulated with LPS. ** *p*  <  0.01; two-tailed Mann–Whitney test.

**Figure 2 cancers-13-02076-f002:**
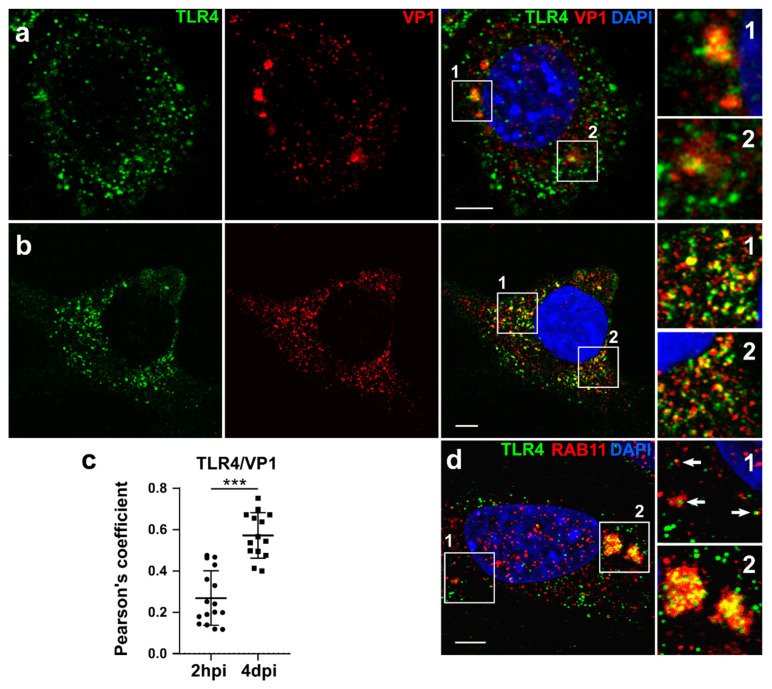
Colocalization of TLR4 and VP1 in MEFs. (**a**) Confocal section of MEFs infected with MPyV at a high virus input (MOI = 10). VP1 (red) and TLR4 (green) were stained by specific antibodies and DNA by DAPI. Colocalization of VP1 and TLR4 was analyzed 2 h post-infection. (**b**) Confocal section of MEFs infected with MPyV at a low virus input (MOI = 1). VP1 (red) and TLR4 (green) were stained by specific antibodies and DNA by DAPI. Colocalization of VP1 and TLR4 was analyzed 4 days post-infection. (**c**) Colocalization between intracellular virus and TLR4, expressed as Pearson′s coefficient. (**d**) Confocal section of MEFs infected with MPyV at a high virus input (MOI = 10). Rab11 (red) and TLR4 (green) were stained by specific antibodies, DNA by DAPI. Colocalization of Rab11 and TLR4 was analyzed 2 h post-infection. White squares show enlarged regions of VP1 and TLR4 or TLR4 and Rab11 colocalizations. Bars = 10 µm. *** *p* < 0.001; Kolmogorov–Smirnov test.

**Figure 3 cancers-13-02076-f003:**
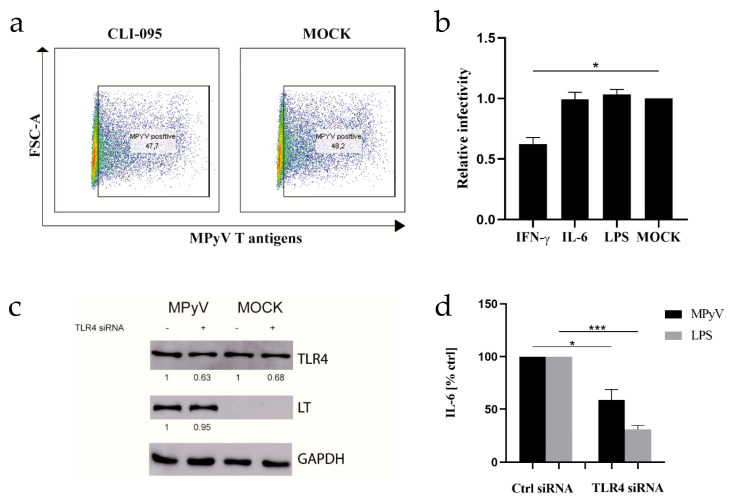
Effect of TLR4 signaling on MPyV lifecycle in MEFs. (**a**) Flow cytometry analysis of the percentage of MPyV-positive MEFs determined by expression of T antigens in the presence or absence (MOCK) of TRL4 inhibitor CLI-095 (10 µM) (**b**) Relative infectivity of MPyV in 3T6 cells pretreated with IFN-γ (100 IU/mL), IL-6 (10 ng/mL), LPS (10 µg/mL) or mock-treated (MOCK) determined by expression of T antigens. Relative infectivity was assessed by flow cytometry, and data were normalized to MOCK. (**c**) The effect of TLR4 silencing on TLR4 and LT levels in MPyV-infected MEFs was followed by Western blot. The values shown below each band represent the relative quantity of TLR4 or LT determined by densitometry normalized to the MOCK-infected MEFs treated with control siRNA. GAPDH was used as a loading control. (**d**) The effect of TLR4 silencing on the IL-6 production in MEFs infected with MPyV or stimulated with LPS measured by ELISA. * *p* < 0.05; *** *p* < 0.001; two-tailed *t*-test.

**Figure 4 cancers-13-02076-f004:**
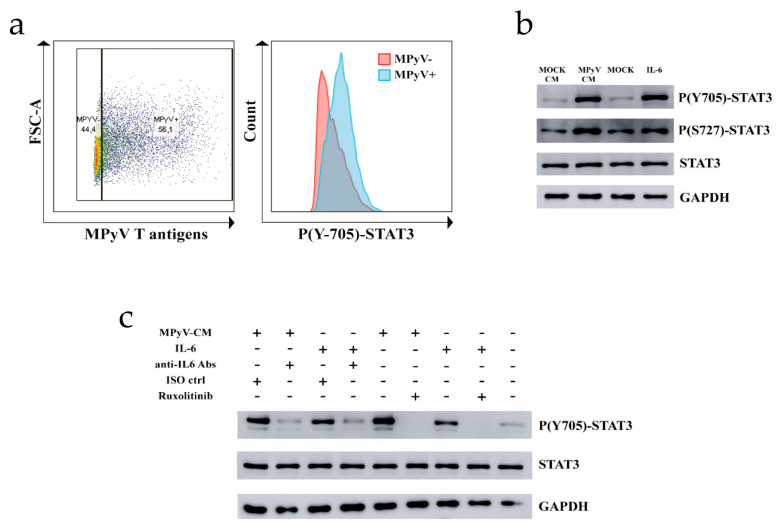
Induction of STAT3 phosphorylation in MEFs by MPyV-CM. (**a**) Fluorescence intensity of STAT3 phosphorylation at Y705 in T-antigen-positive or -negative MEFs. MEFs were infected with MPyV to reach approximately 50% T-antigen-positive cells to assess the distribution of STAT3 phosphorylation in one sample using PhosphoFlow. Red-colored histogram represents T-antigen-negative MEFs, whereas blue-colored histogram represents T-antigen-positive MEFs. (**b**) Western blot analysis of STAT3 phosphorylation at Y705 and S727 in MEFs treated with MPyV-CM or MOCK-CM. Recombinant IL-6 (10 ng/mL) was used as the positive control. Total STAT3 and GAPDH served as the loading control. (**c**) The effect of anti-IL-6 antibodies (Abs) and JAK1/2 inhibitor Ruxolitinib on the MPyV-CM-induced STAT3 phosphorylation at Y705. As a control for anti-IL-6 antibodies, an isotype antibody control (ISO ctrl) was included. The results are representative of three independent experiments.

**Figure 5 cancers-13-02076-f005:**
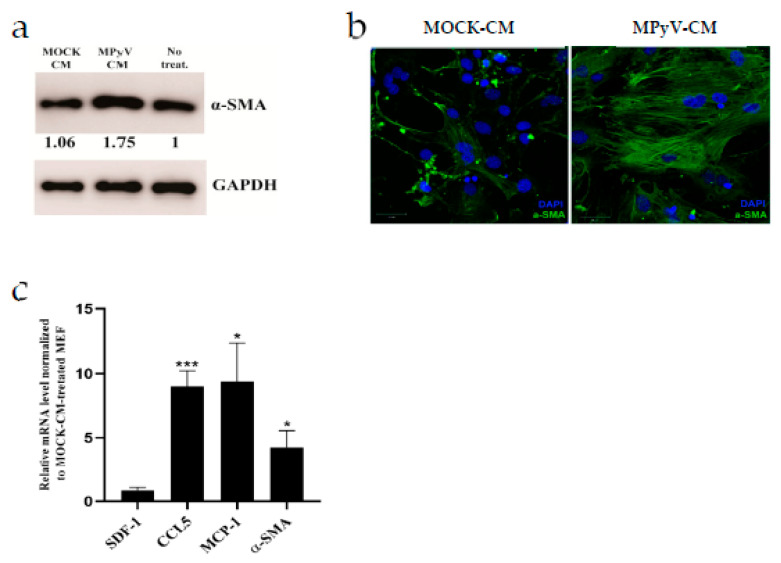
MPyV infection induces a cytokine environment that changes the fibroblast phenotype. (**a**) Analysis of alpha-smooth muscle actin (α-SMA) expression in MEFs treated with MPyV-CM or MOCK-CM. The values shown below each band represent the relative quantity of α-SMA determined by densitometry normalized to non-treated MEF. GAPDH was used as the loading control. The α-SMA expression was analyzed 48 h post-treatment. (**b**) Confocal images of α-SMA distribution in MEFs treated with MPyV-CM or MOCK-CM. Laser settings and acquisition conditions were constant for both samples. The α-SMA distribution was analyzed 48 h post-treatment. (**c**) The qPCR analysis of relative mRNA levels in MEFs treated with MPyV-CM. The qPCR analysis was performed 48 h post-treatment. Relative mRNA level was normalized to MOCK-CM-treated MEFs. Mouse Ppia was used as the endogenous control. The data are the mean ± SEM of four independent experiments. * *p* < 0.05; *** *p* < 0.001; two-tailed *t*-test.

**Figure 6 cancers-13-02076-f006:**
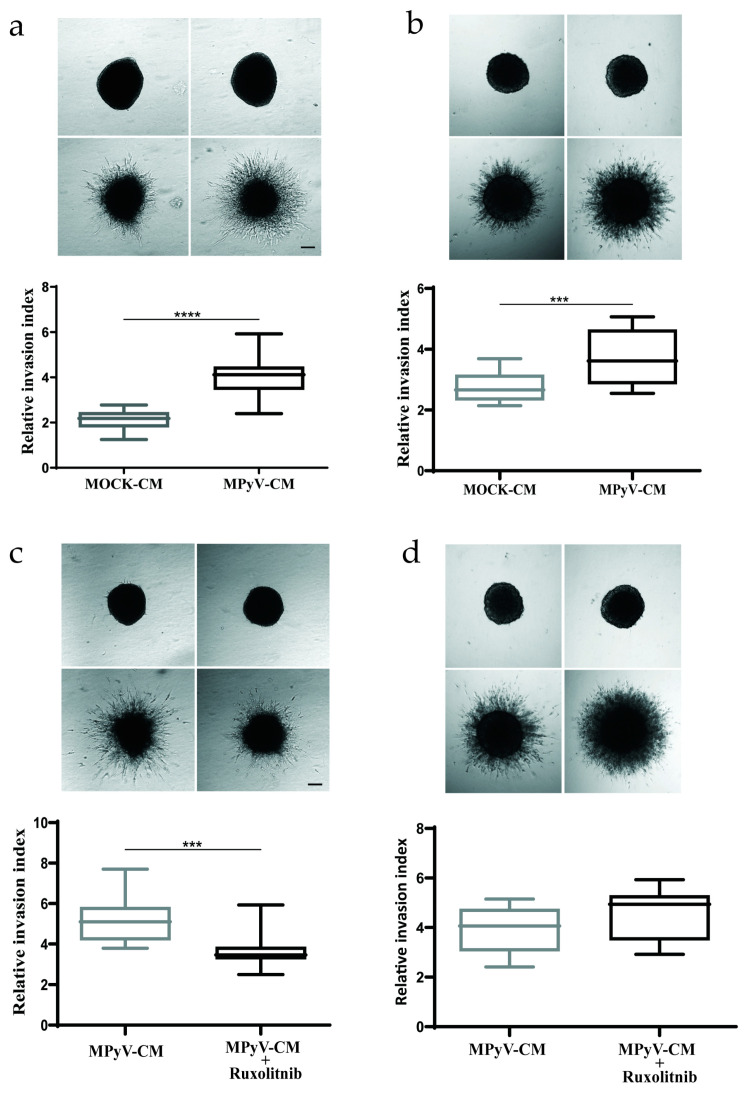
MPyV-induced cytokine environment promotes cell invasiveness. (**a**) Invasiveness of MEF cell-spheroid in a 3D collagen matrix in the presence of MPyV-CM or MOCK-CM. (**b**) Invasiveness of CT26 cell-spheroid in a 3D collagen matrix in the presence of MPyV-CM or MOCK-CM. (**c**) The effect of JAK1/2 inhibitor ruxolitinib on MPyV-CM-mediated MEF spheroid invasiveness a 3D collagen matrix. (**d**) The effect of JAK1/2 inhibitor ruxolitinib on MPyV-CM-mediated CT26 spheroid invasiveness in a 3D collagen matrix. The data are the mean ± SEM of three independent experiments with a minimum of five spheroids per condition. The spheroid invasiveness was analyzed 48 h post-treatment. *** *p* < 0.001; **** *p* < 0.0001; one-way ANOVA test.

## Data Availability

The data presented in this study are available in Appendix A here.

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
