# Peer review of "TLR4-Mediated Recognition of Mouse Polyomavirus Promotes Cancer-Associated Fibroblast-Like Phenotype and Cell Invasiveness"

_cancers, 2021, doi:10.3390/cancers13092076_

Round 1

Reviewer 1 Report

In this study Janovec and authors present the a mechanism of MPyV that through TLR4 induced cytokine environment CAF-like phenotype and increases cell invasiveness. The study is well designed and the experiments are presented in clear and appropriate way. There are few concerns that need to be addressed to further support the study and make it more coherent:

  1. Did the authors addressed the effect of TLR4 pathway exclusively and specifically have an effect in PMyV replication? Did they genetically inactivate TLR4 to check where the virus particle localize and accumulate? Are after siRNA silencing localize in different compartnemtns. The siRNA of TLR4 seems very weak; is there others means that other can address? Stable inactivation by shRNA or Crispr/Cas9?
  2. Where MPyV particles accumulate in endosomes, early/late? Do they localize in autophagosomes? Is ther accumulation of particles when lysosomes and autopahgosomes are blocked? Colocalization in figure 2 needs quantification.
  3. Does the effect of CM on pSTA3 comes from soluble factors or from exosomes and extracellular vesicles? Did the authors treat the cells only with exosome fraction isolated from their supernatant?

Minor comment

The introduction and the discussion are long and sometimes misleading for the study. I would recommend decreasing them accordingly. 

Reviewer 2 Report

In this manuscript,  Janovec et al., demonstrate that TLR4-mdeiated recognition of MPyV induces a cytokine environment that induces a  cancer-associated fibroblast (CAF)-like phenotype in non-infected fibroblasts and increases cell invasiveness. The experiments are performed well and largely support their conclusions. However, a few edits should be made before publication.

  • line 61 - this isn't quite true. It has recently been shown that MCPyV infect fibroblasts (Liu et al., Cell Host Microbe, 2016) and this can recaptulate the whole virus life cycle. This important study should be mentioned.
  • line 371 - the authors only look at IL-6, so can not state 'pro-inflammatory cytokines'
  • Fig 4 - it seems odd that MPyV-CM induces STAT3 Y705 phosphorylation to the same extend as recombinant IL-6. What concentration of IL-6 was added to the experiment? The authors should show the ELISA data and state the concentration of IL-6 in MPyV-CM, as this should be similar to the concentration of recombinant IL-6 used in the experiment.
  • Fig 6 - axis labels and titles are difficult to read and thus the size should be increased.
  • The induction of an IL-6/cytokine rich environment by MPyV is reminiscent of that induced by HPV (Ren et al., Eur J Cancer, 2013; Morgan et al., Plos Pathogens, 2019). It would be of interest to the reader to discuss these studies to demonstrate the relationship of MPyV with similar viruses such as HPV in driving the malignant phenotyp.

Round 2

Reviewer 1 Report

The authors although did not address all my concerns, they have significantly improved the quality of the manuscript.